# Development of Stable Nano-Sized Transfersomes as a Rectal Colloid for Enhanced Delivery of Cannabidiol

**DOI:** 10.3390/pharmaceutics14040703

**Published:** 2022-03-25

**Authors:** Thope Moqejwa, Thashree Marimuthu, Pierre P. D. Kondiah, Yahya E. Choonara

**Affiliations:** Wits Advanced Drug Delivery Platform Research Unit, Department of Pharmacy and Pharmacology, Faculty of Health Sciences, School of Therapeutic Science, University of the Witwatersrand, 7 York Road, Parktown, Johannesburg 2193, South Africa; 1144014@students.wits.ac.za (T.M.); thashree.marimuthu@wits.ac.za (T.M.); pierre.kondiah@wits.ac.za (P.P.D.K.)

**Keywords:** cannabidiol, transfersomes, rectal colloid, nanoencapsulation, permeation, stability

## Abstract

Current cannabidiol (CBD) formulations are challenged with unpredictable release and absorption. Rational design of a rectal colloid delivery system can provide a practical alternative. In this study the inherent physiochemical properties of transferosomes were harnessed for the development of a nano-sized transfersomes to yield more stable release, absorption, and bioavailability of CBD as a rectal colloid. Transfersomes composed of soya lecithin, cholesterol, and polysorbate 80 were synthesized via thin film evaporation and characterized for size, entrapment efficiency (%), morphology, CBD release, ex vivo permeation, and physicochemical stability. The optimized formulation for rectal delivery entrapped up to 80.0 ± 0.077% of CBD with a hydrodynamic particle size of 130 nm, a PDI value of 0.285, and zeta potential of −15.97 mV. The morphological investigation via SEM and TEM revealed that the transfersomes were spherical and unilamellar vesicles coinciding with the enhanced ex vivo permeation across the excised rat colorectal membrane. Furthermore, transfersomes improved the stability of the encapsulated CBD for up to 6 months at room temperature and showed significant promise that the transfersomes promoted rectal tissue permeation with superior stability and afforded tunable release kinetics of CBD as a botanical therapeutic with inherent poor bioavailability.

## 1. Introduction

Colloidal-based drug carriers provide many advantages for the delivery of bioactives due to their intrinsic physiochemical properties. Colloidal vectors can be easily modified for the site-specific delivery of a wide variety of small molecules. Cannabidiol (CBD), a candidate small molecule constitutes the second most active compound of cannabis and has no evidential psychoactive side effects such as addiction and hallucination [1]. While a growing body of research indicates that the CBD molecule promises a host of pharmacological effects such as anticancer, analgesic, anti-anxiety, or anti-inflammatory properties there are only a few medicinal-grade products in use [2]. Although CBD may have a high potency, its efficacy is limited by poor oral bioavailability ranging between 13 and 19% [3,4,5] due to rapid metabolism, gastric instability poor chemical stability (photo- and thermo-sensitive), and low water solubility, resulting in reduced and unpredictable absorption [3,6]. Therefore, alternative strategies to the development of more effective CBD-loaded drug delivery systems are needed to realize the beneficial pharmacological effects of the CBD molecule by overcoming its biopharmaceutical shortcomings.

Various alternate routes of administration for the delivery of CBD have been investigated in an attempt to maximize its potential therapeutic effects and include but not limited to intranasal [7], pulmonary [8], oromucosal [9], and transdermal [10]. Although intranasal delivery of CBD offers a non-invasive approach to reach a rapid onset of action, the potential for mucosal and respiratory tissue irritation coupled with dosing variability limits this route of administration. Studies have also shown that improvement in the pharmacokinetics of CBD can be achieved by oromucosal delivery for systemic absorption; however, absorption may not necessarily occur via the transmucosal route [11]. Sufficient mucosal residual time coupled with an approach to prevent saliva washout is critical to ensure delivery efficacy via this route [12]. Despite the potential of using transdermal patches in certain applications, this route is limited by the high cost to efficacy ratio relative to other delivery systems for CBD [10]. Hence, there is much impetus for investigating alternative routes of administering CBD based on its purported therapeutic applications.

To the best of our knowledge there are very limited studies that explored the delivery of CBD via the rectal route as a useful approach for poorly bioavailable compounds such as CBD [13,14]. Transmucosal delivery such as via the rectal route is increasingly becoming of interest for patients specifically under palliative care [15]. Rectal administration can provide local action as well as systemic absorption of drugs that can benefit specific patient groups for example in dysphagia/aphagia or where the absorption of sensitive bioactives is required independent of food effects and/or contact with GIT fluids must be avoided [16]. Furthermore, liver metabolism is significantly minimized via the rectal route resulting in superior bioavailability. For example, studies have shown that the bioavailability of drugs such as tizanidine HCl improved 2.18-fold when administered rectally compared with oral administration [17] and highly lipophilic drugs can avoid hepatic metabolism via lymphatic drainage [18]. However, the rectal route also has limitations and one of the major concerns is to overcome the limitation of incomplete or irregular absorption from the rectum primarily due to the significantly smaller surface area available for absorption and other key physiological considerations such as the presence of feces and microbial degradation of bioactives [16]. 

The rectal mucous layer acts as a barrier for drug absorption and depending on the physiochemical properties of the drug absorption can occur via the intra- or inter-cellular route [19]. A significant number of drugs delivered as a rectal dosage form are characterized as small molecules <500 g/mol with high log *p* values > 2.0 [20]. While drug particle size of 50–100 μm is considered suitable for systemic absorption of lipophilic drugs; smaller particle sizes can promote faster rectal absorption [19]. Although there are limited pharmacokinetics studies that specifically compare the bioavailability between the rectal and oral administration, evidence from a two-patient clinical study showed that a rectal cannanbinoid formulation increased the systemic bioavailability relative to an orally administered formulation [21].

Strategies to achieve more predictable and optimized drug absorption from the rectum such as adapting nanotechnology are therefore needed to ensure that this alternate route of administration remains viable for a wider variety of bioactives. Moreover, nano-formulations owing to their ability to reduce systemic toxicity through targeting coupled with potential to serve as diagnostic nanoplatforms can be used to manage colonic diseases such as colorectal cancer (CRC) [22]. Specifically, lipid-based nanocarriers are one of the ideal candidates, as they offer diverse but controlled particle size, allow for controlled release, and can be functionalized with active targeting agents [23]. Rectal administration is not the most preferred route for the management of CRC. In this regard, Wang and colleagues established a proof-of-concept study, where a lipid-based nanoparticle rectal delivery system mitigated in vitro and orthotopic murine colorectal cancer growth [24]. As such, lipid nanocarriers can be seen as highly advantageous for the development of rectal drug delivery systems. Therefore, this study explored the use of nano-sized transferosomes assembled as a rectal colloid to achieve preferential and more stable bioactive absorption via the rectal route for the model botanical therapeutic CBD. Transfersomes are biocompatible ultra-thin elasticized nanocarriers that are similar to (nano)liposomes [25,26]. Liposomes can transport drug through biological barriers by surface adsorption and drug transfer from the vesicles, fusion with the lipid matrix of the barrier for increased drug partitioning, and lipid exchange with cell membranes to facilitate drug diffusion. However, the challenge with liposomes is that their depth of penetration is limited resulting in poor drug concentrations reaching the systemic circulation. 

Transfersomes have therefore been explored to encapsulate hydrophilic, lipophilic, and amphiphilic drugs of low to high molecular weight mainly via the transdermal route [27] with superior bioavailability and stability [28,29]. Compared with (nano)liposomes, transferosomes have a lipophilic elastic nature that can facilitate the intercellular transport of bioactives more deeply and across biological barriers such as the rectum for enhanced permeation and absorption [26]. Transfersomes comprise phospholipids with an aqueous core structure and an outer complex lipid bilayer [26,30]. Incorporation of an edge activator provides the unique elasticity that is needed for self-optimizing deformability of the transfersome structure during permeation across varying pore sizes [29]. Edge activators are bilayered softeners constituting single-chain surfactants such as sorbitan esters, sodium cholate, and polysorbates that reduce interfacial tension and destabilize lipid bilayers of vesicles to facilitate their deformation for easier intercellular transit. [27,29]. Herein we describe the exploitation of the physicochemical properties of transferosomes in the development of a rectal colloid to achieve more stable release, absorption, and bioavailability of CBD. Detailed in vitro characterization and ex vivo studies were undertaken to support the conclusions that demonstrate the rectal route may be a viable route for the stable delivery of CBD with optimum release and absorption kinetics.

## 2. Materials and Methods

CBD 1.0 mg/mL in methanol standard, soybean lecithin (SL), polysorbate 80, cholesterol, methanol, chloroform, phosphate-buffered saline (PBS) tablets, Triton X-100, hydrocortisone, and mannitol. Millipore^®^ were sourced from Sigma-Aldrich (St. Louis, MO, USA). High-pressure liquid chromatography (HPLC) column (4.6 mm × 150 mm: 3.7 µm) and membrane filters (0.22 µm) where procured from Waters Corp. (Billerica, MA, USA). Cannabidiol (CBD) isolate was gifted. The colorectal tissue was acquired from University of Witwatersrand Central animal service (CAS). High purity solvents were used for instrumental analysis and other reagents employed in the study met analytical standards.

### 2.1. Development of the CBD-Loaded Transfersome Based-Rectal Colloid Using Thin Film Hydration

An adapted thin layer hydration method with varying lipid ratios was used to synthesize transfersomes for optimal CBD encapsulation, (Figure 1) [28]. The lipid phase was prepared using a blend of soya lecithin, polysorbate 80, cholesterol, and 50 mg CBD. The lipid phase was then dissolved in an organic solvent containing chloroform and methanol at a 2:1 ratio (1). The contents were subjected to magnetic stirring at room temperature at 100 rpm until a homogenous mixture was achieved. The removal of organic solvent was achieved using a rotary vapor in a water bath (40 °C; 100 rpm) for 5 min until the thin lipid film was formed (2). Further removal of remaining organic solvent was achieved by placing the thin lipid film in a fume hood at room temperature for a two-hour period. The films were thereafter rehydrated using PBS (pH: 7.4 under vigorous stirring; volume: 15 mL) (3), and then left for two hours which yielded multilamellar vesicles (MLVs) (4). The resultant MLVs were placed into sonicator bath (Sonics Vibra Cell, Newtown, CT, USA) at ambient temperature for 15 min to produce unilamellar vesicles (5). To obtain uniform size distribution, unilamellar vesicles were gradually filtered through a membrane filter (0.22 μm) [31]. The resulting transfersomes were frozen at −80 (6) before lyophilization for 16 h at −60 °C (7) was carried out to prevent hydrolysis during storage. Mannitol (15%) was used as a cryoprotectant [32]. The composition of transfersomes is presented in Table 1.

### 2.2. Determination of Morphology, Particle Size and Zeta Potential of the CBD-Loaded Transfersomes

Sample preparation for transmission electron microscopy FEI Tecnai T12, (FEI Company, Hillsboro, OR, USA), briefly involved the preparation of a diluted CBD-loaded transferosomal dispersion with distilled water (1:10) followed by sonication for 5 min with a bath sonicator (Sonics Vibra Cell, CT, USA). Thereafter a drop of dispersion was allowed to adsorb onto a carbon-coated copper grid. Extra dispersion was carefully cleared with a filter paper and the grid was washed twice with deionized water for 3–5 s before the uranyl acetate 2% aqueous solution stain was added [28]. The sample was air dried overnight. The transfersomes were examined for size and shape using 6 kV scanning electron microscope (SEM) (Jeol JSM-120, Tokyo, Japan) operating under high vacuum [33]. For SEM analysis, sufficient amount of sample was placed on an aluminum stub, air dried and coated with gold-palladium. 

Dynamic light scattering measurements were performed on a 1:10 dilution of transfersomal suspension: deionized mixture [34]. The diluted samples were sonicated for five minutes and approximately 4 mL of the sonicated sample was poured into disposal cuvettes (quartz cuvettes) to measure the average particle size and polydispersity index (PDI), while folded capillary cells were used to quantify zeta potential using zeta sizer NanoZS instrument (Malvern Instruments (Pty) Ltd., Worcestershire, UK). Three replicates (*n* = 3) of each sample were analyzed and the mean average for each sample was reported accordingly. For the analysis of CBD, a 20 mg/mL CBD in ethanol solution was prepared and a 1:10 dilution was carried out with PBS (7.4).

#### 2.2.1. Determination of CBD Entrapment Efficiency (%EE)

Quantification of amount of CDB was conducted by the high-performance liquid chromatography (HPLC). Isocratic elution was carried out using a mobile phase of methanol: distilled water 85:15% at a flow of rate of 1 mL/min. A reversed phase Symmetry^®^ C18 column (pore size; 3.5 µm; dimensions: 4.6 × 150 mm) was employed for the procedure. A 20 µL sample aliquot was injected and measured at 220 nm. Cannabidiol had a retention duration of 5.6 min, normalization was done with hydrocortisone, which had a retention time of 2.6 min, and the overall run time was 10 min. To determine the quantity of CBD entrapped, 5 mg of lyophilized transfersomes were disintegrated in Triton X-100 (2 mL, 1% *v*/*v*) diluted with PBS; and subjected to stirring for 2 h over magnetic stirrer to facilitate the drug. The resulting mixture was filtered with a polycarbonate membrane filter (0.2 µm). A 0.5 mL aliquot was further diluted with methanol (0.5 mL) and analyzed using HPLC [29]. Once the drug amount was quantified the following equations where employed (Equation (1)) to give %EE as per reported methods [35,36].
(1)$$\mathrm{Entrapment}\;\mathrm{efficiency}\;\left(\%\right)=\frac{\mathrm{Mass}\;\mathrm{of}\;\mathrm{CBD}\;\mathrm{encapsulated}}{\mathrm{Mass}\;\mathrm{of}\;\mathrm{CBD}\;\mathrm{added}\;\mathrm{to}\;\mathrm{formulation}}\times 100$$

#### 2.2.2. Determination of Excipient Compatibility via Thermal and Spectroscopic Analysis

Differential scanning calorimetry was employed for the thermal characterizations of mannitol (cryoprotectant), pure CBD, T3, and T6 (Mettler Toledo, DSC1, STARe System, Swchwerzenback, Switzerland). Samples weighing between ~10 mg were crimped in perforated aluminum pans and heated under an inert nitrogen environment. A scanning rate of 10 °C/min was employed, with a −10 °C insertion temperature, and a scanning temperature range of −10−300 °C. Tests were conducted in triplicates for each sample [3,37,38]. Thermogravimetric analysis was performed on pure CBD, T3, and T6 to evaluate thermal degradation properties (PerkinElmer, TGA 4000, Llantrisant, Wales, UK). A suitable sample quantity was deposited in the crucible and samples acquired from 30–900 °C (10 °C/min), under inert conditions. The Pyris™ program was used to analyze the data. The FT-IR spectra of mannitol (cryoprotectant pure CBD, T3, and T6) was recorded using a PerkinElmer^®^ Spectrum Series (PerkinElmer Ltd., Beaconsfield, UK). The method, adapted from literature was modified as follows: appropriate sample quantities were placed on a diamond stage and FT-IR measurements were performed between a range of 650–4000 cm^−1^.

### 2.3. Kinetics Evaluation of the Release Study of CBD from the Transferosomes

In a typical experiment, CBD-loaded transfersomes (T4 to T6) were immersed in phosphate-buffered saline (PBS) (10 mL, pH 7.4), to which polysorbate 80 (2% *w*/*v*) was added and the samples were incubated in an orbital shaker (37 ± 0.5 °C) and subjected to mechanical stirring at 50 rpm for 7 h. Aliquots of 1 mL of the supernatant were removed at pre-determined time points and replenished with equal quantity of buffer. The sample drawn was analyzed for the amount of CBD using the HPLC method described above, following filtration using a membrane filter (0.2 µm). A calibration curve, using the method of internal standard addition was constructed over a range of 1–100 μg/mL and was used to quantify the amount of CBD. The linear regression (and *r*^2^ = 0.9995) was within acceptable limits as per previously validated RP-HPLC method reported [39].

The kinetics of cumulative drug release from transfersomes was studied utilizing zero order, first order, Higuchi model, and Koresmeyer–Peppas mathematical models. DDSolver, a Microsoft Excel data analysis tool, was used to determine the degree of correlation. Based on the highest correlation, the best-fitting mathematical model was chosen.

Similar mathematic approach has been used previously by El-Gizawy and co-workers [40]. The equations used in this study were as follows:*Q* = *K*_0_
*t* for zero order(2)
*In* (*Q_t_*) = *In*(*Q*_0_) + *K*_1_*t* for first order(3)
*Q* = *K_H_ t*^1/2^ for Higuchi model(4)
*Q = k·t^n^* for Koresmeyer-Peppas model(5)
where *Q* is the cumulative amount of medication released at time *t*, *t* denotes the time in minutes, and *K*_0_ denotes the zero order rate constant. *K*_1_ denotes the first order constant. Higuchi’s dissolution constant is *K_H_*. *K* is the constant that integrates the dose form’s structural and geometric features. *n* stands for the release exponent.

### 2.4. Determination of Ex Vivo Permeation of CBD-Loaded Transferosome

Ethics waiver was obtained from Wits central animal service (CAS) to collect colorectal tissue of Sprague Dawley rat weighing between 200 and 300 g. The tissue was stored in the formalin after collection. It was then submerged in the PBS (7.4) maintained at 37 °C, and used to predict permeation. Franz diffusion cell with a diffusional area of 1.77 cm^2^ was used to assess permeability of CBD and T6. The receptor chamber with the capacity of 12 mL was filled with PBS (pH 7.4; 37 °C) with 2% *w*/*v* polysorbate 80. The temperature was maintained at 37 °C via a heat exchange jacket (simulating rectal temperature). The colorectal tissue, with a thickness of 0.056 cm, was sandwiched between the donor compartment and receptor chamber. The CBD isolate (3 mg) and T6 (300 mg) were weighed and distributed in 3 mL of PBS (pH 7.4; 37 °C) and uniformly spread on the membrane sandwiched between donor and receptor compartments. Samples (1 mL) were removed from the receiving compartment medium at fixed time points over a 7 h period. Fresh buffer was replenished with the drawn volume of samples. The samples were quantified for CBD using the HPLC method described Section 2.2.1. The flux was calculated to determine the rate of permeation.

#### 2.4.1. Colorectal Membrane Integrity Assessment

Colorectal membrane integrity was assessed using FT-IR and ionic conductivity measurements before and after exposure to Cannabidiol and T6. Prior to and after CBD and T6 exposure, ionic conductivity was measured on a Seven Multi S40 pH/electrical conductivity meter (Mettler–Toledo, Greifensee, Switzerland) [41]. Colorectal tissue structural integrity before and after exposure of CBD and T6 was analyzed using FTIR spectroscopy (PerkinElmer, UK).

#### 2.4.2. Determination of the Physical Stability of the CBD-Loaded Transfersome

For determination of stability, method by Wu et al., 2019 was adapted with modification. Lyophilized transfersomes were placed at ambient temperature in a dark room for six months to assess stability. Stability was evaluated at 0, 3, and 6 months. Lyophilized transfersomes were dispersed in distilled water and stability was assessed based on particle size, PDI, zeta potential [42]. Physical appearance such as color change was also observed.

## 3. Results and Discussion

In this study, only the influence on the amount of surfactant for the preparation of the CBD-loaded transferosomes was investigated. To infer changes, CBD-loaded transfersomes were investigated for different physicochemical characterizations which included particle size, potential, and in vitro CBD release. The optimal CBD-loaded transfersome formulation was selected based on EE%, ideal vesicular size, and high degree of drug amount released (low retention of CBD). Additional studies of the optimized formulation were carried out in terms of stability of formulation, morphology analysis, and permeation across the excised colorectal membrane.

Phospholipids are directing agents for the formation of vesicles. Soybean lecithin was selected as the phospholipid because it has been successfully employed in previous studies for the entrapment of lipophilic drug molecules with good encapsulation capacity and offered controlled drug release [43]. This selection is not limited to the favorable drug loading properties of soybean lecithin, but also is based on safety and relative cost effectiveness of this lipid which would be beneficial to large-scale preparation of transfersomes. Edge activators contribute to the deformable properties of transfersomes. Polysorbate 80 was selected as the edge activator because it affords smaller vesicle size that is also suitable for rapid rectal absorption [17]. Through the addition of Polysorbate 80, it was possible to obtain small and uniform particle size, while enabling increased CBD release which is suitable for rectal absorption. Cholesterol was included to strengthen the membrane of the transfersomes. A ratio of soy lecithin to cholesterol of 2:1 was selected based on previous report that indicated that this ratio gave the best encapsulation efficiency [44]. Phosphate buffer (7.4) served as a buffering agent and hydrating medium suitable for rectal administration. Triton X 100 was employed to break down the transfersomes to enable the accurate measurement of the total amount of drug entrapped as previously described by Moawad and colleagues [17].

### 3.1. Visualisation of Vesicular Shape and Surface Morphology

The size and morphology of optimal transferosomal dispersion (T6) were viewed on electron microscope (80 kV). The TEM images revealed spherical vesicles, with unilamellar structure (Figure 2A) which were similar to previously reported TEM images of transfersomes with uranyl acetate staining [45]. Using Image J software, the average diameter was found to be 87.31 ± 12.62 nm. As the DLS measurements are conducted on solvated transferosomes, the mean diameter was expected to be slightly smaller for the TEM analysis and this trend was congruent to previous studies [45].

The surface morphology was viewed via SEM imaging which also depicted spherically shaped transfersomes (Figure 2B). The rectal colloids were well dispersed with no visible phase separation as evident in Figure 2C and the formulations were stored as a dry lyophilized powder (Figure 2D) respectively.

### 3.2. Assessment of Hermal Stabiilty in Pristine CBD and CBD-Loaded Transferosomes

Pure CBD, T3, and T6 were evaluated for physical form and degradation. Figure 3A represents DSC thermogram. A sharp endothermic peak was observed at 60–75 °C for CBD corresponding to its melting point, as supported by literature [3,46]. In T3 and T6 formulations, a sharp endothermic peak was observed at 130–150 °C, with a second sharp endothermic peak recorded at 150–169 °C, confirming the presence of mannitol as it was added as a cryoprotectant and maybe present in different crystalline forms after freeze drying. The thermal events of formulation T3 are similar to that of formulation T6, with the absence of the CBD endothermic peaks confirming successful loading of CBD into the transfersomes [47]. The absence of the characteristic CBD peak can be attributed to the embedding of CBD within the lipid bilayer of the transfersomes. Furthermore, the drug in transfersomes is in an amorphous phase and may possibly be homogenously distributed in the transfersomes, hence resulting the absence of the characteristic CBD peak [47]. The degradation of CBD was analyzed using TGA. The TGA thermogram demonstrated that CBD significantly started losing its mass from the temperature of approximately 225 °C and lost ~100% of its mass at 340 °C. The empty transfersomes (T3) and CBD-loaded transfersomes colloid (T6) started losing their mass at 290 °C and lost ~100% of the mass at 370 °C. The temperature at which transfersomes-loaded CBD started degradation is significantly higher compared to that of the free CBD. There is no thermal variation between the empty and CBD-loaded transfersomes colloid demonstrating that CBD has been loaded within the lipid bilayer hydrophobic tails. This implies that the thermal stability of CBD was improved after encapsulation of CBD into transferosomes (Figure 3B). These thermal results are similar to the reported thermal properties of a CBD-loaded polymeric film [48].

### 3.3. Comparative Analysis of the Molecular Vibrations of CBD and CBD-Loaded Transferosomes

FT-IR spectra of pure CBD, T3, and T6 is represented in Figure 4. FT-IR provides insight into the molecular interaction between drugs and excipients. In the FT-IR spectrum of CBD (Figure 4 (A)), significant molecular vibrations in the region of 3401 to 3513 cm^−1^ corresponding to the O-H (aromatic) stretching vibrations while the bands at ~3000 cm^−1^ were assigned to C-H stretching (phenyl), ~2914 cm^−1^ was indicative of methyl and methylene groups, ~1575 cm^−1^ was indicative of C=C stretching (phenyl ring), and C-O stretching vibrations were at ~1216 cm^−1^ [49]. The FT-IR spectra of formulation T3 (Figure 4 (B)) and T6 (Figure 4 (C)), depicted a broad band in the region of 3400–3000 cm^−1^, which corresponds to the hydroxyl groups (O-H stretch). The sharp peak at 2935 cm^−1^ corresponds to the alkyl C-H stretch [50]. Absence of the characteristic CBD peaks ~1575, ~1216 cm^−1^ in the transfersomes formulations indicate a drug-excipients interaction, possibly due mechanical encapsulation of CBD within the transfersomes [46]. Furthermore, the lack of CBD bands in the T6 spectrum may stipulate that significant amount of CBD lies embedded within the lipid bilayer relative to the surface of the transfersomes [46].

### 3.4. Evaluation of Particle Size, Zeta Potential, and Entrapment Efficiency

The mean particle size of empty transfersomes colloidal suspension ranged from 73.2 ± 0.065–86.8 ± 0.76 nm and the CBD-loaded transfersomes colloidal suspension size ranged from 102.2 ± 0.72 to 130.1 ± 0.64 nm respectively as shown in Table 2. The mean particle size of CBD ethanolic solution-diluted phosphate buffer (7.4) was relatively larger. All formulations prepared were in nano sized range. However, the size of CBD-loaded transfersomes was slightly larger than of the empty transfersomes which was expected due to encapsulation of CBD. The particle size of lyophilized empty and CBD loaded transfersomal colloid formulation slightly increased to 92.5 ± 0.77–101.5 ± 0.20 nm and 121.7 ± 2.11–146.6 ± 0.23 nm respectively. The size of CBD-loaded transfersomes was slightly larger than of the empty transfersomes which was expected due to encapsulation of CBD. The change in particle size after lyophilization agreed with previous studies where a lipophilic drug was incorporated in transfersomes prepared with SPAN 80 and polysorbate 80 respectively [51]. Nonetheless, the obtained size of the CBD transfersomal formulations is suitable for enhanced permeation of CBD by enabling CBD to cross sufficiently through the pores of biological membranes. The PDI for empty transfersomes ranged from 0.24–0.30 and CBD-loaded transfersomes ranges from 0.26–0.29 respectively. These values were satisfactory as the values where close to 0.3 which is acceptable for nano phospholipid structures [52]. Furthermore, these values revealed that the transfersomal formulations showed a narrow distribution and uniformity in particle size. The formulations with the smallest size and PDI for empty transfersomes and CBD-loaded transfersomes were T1 and T4 respectively with the lowest amount of polysorbate 80 added. Increase in edge activator (polysorbate 80) did not result in an associated decrease in particle size. Other studies also found that polysorbate 80 had negligible effects on size and PDI [53].

The zeta potential of transfersomes was studied to assess the surface characteristic such as stability of formulated transfersomes. Stability is affected by surface charge of vesicles, as the surface charge contributes to a repulsive energy between particles, preventing transfersomes aggregation [53]. The T3 and T6 formulations demonstrated zeta potentials of −20.8 ± 0.95 mV and −15.97 ± 1.3 mV respectively. The zeta potential of the lyophilized transfersomes both empty and CBD loaded was more negative within the range of −21 to −30 mV. The more negative zeta potential after lyophilization demonstrates that the lyophilization of the transfersomes enhanced its stability.

The negative potential in the aqueous test solution was most likely due to the adsorption of the OH^−^ ions on the soy lecithin lipid chains [54]. Reported studies have shown that the charge of the lipids and surfactant contributes in an additive manner to the overall electrostatic charge on the vesicular surfaces [55]. The zeta potential of the pristine CBD in phosphate buffer (7.4) was very small and is indicative of neutral molecules [56]. As CBD would be incorporated between the hydrophobic tails of the lipid bilayer membrane, integration of CBD into the lipid bilayer membrane may have destabilized the transfersomes, hence there was a reduction in the zeta potential on the CBD-loaded transfersomes relative to the free transfersomes. Notably, relative to T1 and T2, T3 depicted a more negative potential which is linked to the higher amount of edge activator used in the T3 formulation.

There was a direct relationship between amount of polysorbate 80 added and the entrapment efficiency. Increase in amount of polysorbate 80 added resulted in increased entrapment efficiency, with the highest entrapment efficiency of 80% demonstrated with the addition 100 mg of polysorbate 80. This effect is likely caused by the size of the fatty acid chains of polysorbate 80 and its hydrophilic/lipophilic balance (HLB). For instance, the loading of a lipophilic drug is favored by formulating with a low HLB based surfactant. It has also been shown that at higher concentrations of surfactants, there are higher formation of vesicles which can allow for more space in the hydrophobic lipid layers which enables higher entrapment of the hydrophobic drug molecules [27]. Refer to Appendix A for zeta size and potential data.

### 3.5. Assessment of the In Vitro Release of CBD from CBD-Loaded Transfersomes

Drug release from CBD-loaded transfersomes (T4 to T6) were assessed over a seven-hour time at 37 °C, represented graphically in Figure 5. The release was completed at 7 h with approximately 95% of the drug was released for formulations T4 and T6, while approximately 80% of the drug was released for formulation T5. An initial rapid release was observed for T4, T6, and T5. Similar patterns of release have been reported for an encapsulated hydrophobic drug where an initial rapid release preceded a slow-release phase via proposed drug diffusion from the lipid bilayer [57].

Notably T4 and T6 showed quick CBD release. Higher surfactant ratios have been reported to give quick drug release as the surfactant facilitates drug solubility in aqueous media [40]. However, the release profile for T5 showed a relative hindrance to drug release which could be due to the greater ordering of the lipid bilayers endowed by the surfactant at the ratio used. Nonetheless at very high surfactant ratios the release pattern of T6 is similar to T4.

The release of the CBD from the T5 formulation was significantly (*p* = 0.01; *p* < 0.05) less in comparison to T6, while there was no significant difference for T4 and T6 (*p* = 0.20). Furthermore, there is a notable relationship between the particle size and the release of CBD. The transfersomes colloidal formulation with the smallest particle size (T4) resulted in rapid rate of drug release, followed by T5 then T6. The impact of smaller particle size of transfersomes colloidal formulation was most likely due to the large surface area of smaller particles which leads to a short diffusion distance for the released CBD [55]. However, T6 has slightly increased release as compared to T5 which is contrary to what was expected. The drug release of T6 was expected to be slower compared to T5, however as the quantity of the surfactant (polysorbate 80) increased to the maximum of 100 mg, it affected the membrane permeability and rigidity of transfersomes because of the nature of polysorbate 80. Polysorbate 80 comprises lipophilic tails with long and unsaturated (C18) moieties facilitating the incorporation of CBD within the lipid bilayer and resulting in more permeable vesicle membrane [58]. Therefore, for T6 the release is likely due to the fact that at the prescribed % composition of the surfactant, the polysorbate 80 molecules are associated with the lipid bilayer and this facilitates the partition of CBD into the release medium [59]. For T5, it is possible that the ratio of CBD to vesicle number exceeds the vesicular loading capacity (%EE for T5 close to %EE for T6) and this could have led to premature (initial) leakage of CBD from the lipid bilayers [58].

The drug release study proved that the transfersome formulation depending on the amount of edge activator can sustain the drug release due to their reservoir effects [28]. The drug release results were fitted into drug release kinetics to determine the best fitted mathematical model as shown in Table 3. T5 demonstrated zero order drug release, while T4 and T6 formulations demonstrated Higuchi mode, based on correlation coefficient value (R^2^). Zero order implies that the drug release per unit time is not influenced by drug concentration added, meaning a constant amount of drug is being released. The Higuchi model revealed that drug penetration mechanism is by diffusion indicating controlled release. Furthermore, the Korsmeyer–Peppas model supported these findings, with *n* values achieved for T4–T6 being less than 0.5, which is classified as Fickian diffusion. Fickian diffusion occurs when the release is rapid at first and gradually reduced over time. The release mechanism is diffusion driven by chemical gradient [60].

### 3.6. Evaluation of the Comparative Ex Vivo Permeation of CBD and CBD-Loaded Transfersomes

Ex vivo permeation studies provide vital information about the movement of the drug across the cell membranes into the blood stream. The mechanism of permeation/absorption from the rectum is transcellular passage across the cell membrane [17]. Based on drug release study, physiochemical properties and entrapment efficiency, T6 was deemed the most suitable formulation for rectal CBD administration and was thus used for permeation studies. The permeation properties of formulation T6 was assessed for a 5-h duration. The cumulative flux was used to compare permeation of transfersomes and CBD. The transfersomes had fully permeated through the colorectal membrane within an hour, with the flux of approximately 1.7 mg/cm^2^/h as compared to the 1 mg/cm^2^/h of the CBD alone, as shown in Figure 6. The amount of CBD permeated from T6 formulation was moderately (*p* = 0.006; *p* < 0.05) more compared to CBD. Transfersomes displayed relative higher permeation than the drug alone attributed to their ultra-deformable elastic and flexible nature that allows them to squeeze through smaller pores, these results are in alignment with the other studies [17]. Furthermore, transfersomes had smaller size and negatively charged. Smaller size increases surface area for absorption. CBD has significantly large particle size as shown in Table 2, which may have contributed to its delayed permeation or absorption. Membrane integrity studies were performed to ensure that the membrane remained intact before and after exposure of CBD and T6. Ionic conductivity results of before and after exposure of CBD and T6 had insignificant differences with the conductivity of 258.3 uS/cm before the exposure and 257 uS/cm after the exposure, confirming that the membrane remained intact. Ionic conductivity results of before and after exposure of CBD and T6 had insignificant differences confirming that the membrane remained intact. Furthermore, relative to Figure 7 (A) the colorectal membrane after exposure of CBD and T6 remained intact (Figure 7 (B)). This is evident as there were no noticeable changes in the peaks at 2923 and 2853 cm^−1^ which illustrated that the membrane remained intact during the study [61].

### 3.7. Assessment of the Physicochemical Stability of the Transfersomes

The physicochemical stability of lyophilized transfersomes was assessed over a 6-month period, at ambient room temperature (25 °C) and humidity, in the absence of light for 0, 3, and 6 months. Figure 8 represents stability of transfersomes evaluated based on size, PDI, and zeta potential for over six months. There was insignificant increase of size from 146.63 ± 0.23 to 148.00 ± 2.00, aligning with the findings of other studies [62]. There was negligible decrease in PDI from 0.25 ± 0.0056 to 0.23 ± 0.0081. Zeta potential slightly decreased from −29.13 ± 1.3 to −27.91 ± 0.4 after six months. However, these results still prove that lyophilized transfersomes improved CBD stability because the decrease in zeta potential was negligible. Furthermore, to ascertain stability in terms of CBD content, the entrapment efficiency% was found to be 80.10% ± 0.25 and 78.51% ± 0.22 for month zero and month 6 respectively. This result proves that the transfersomes can enhance stability of CBD for a period of 6 months. No change in color was observed, the lyophilized transfersomes remained cream white for six months as depicted in Figure 9. From these results it can be inferred that the transfersomal formulation can maintain stability of CBD for about six months.

## 4. Conclusions

The current investigation aimed to assess the potential use of transfersomes as an encapsulation strategy for cannabidiol for rectal drug delivery. Based on this approach, the solvent, lipids, edge activator, and hydrating medium were selected for suitability for rectal administration. The impact on the surfactant concentration was investigated in terms of particle size, zeta potential, and %EE for the formulations preprepared. The effectiveness of this proposed method has been substantiated by the collection of in vitro release data and ex vivo permeation data across the colorectal membrane. Nano-sized CBD-loaded transferosomes were successfully synthesized and assembled into a rectal colloid to enhance the rectal absorption of CBD. Results indicated that stable transfersomes with average particle size of 102.2–130.1 nm were able to efficiently entrap 55.7–80.0% of CBD with varying compositions of the edge activator that facilitated relaxation of the lipid bilayer for superior encapsulation of CBD. Lyophilization of the transferosomes preserved and extended its physicochemical stability over a period of 6 months. Nonetheless, future research should fully examine the stability of CBD in the formulations under accelerated testing conditions as per ICH guidelines. Ex vivo permeation studies revealed that transfersomes considerably improved the diffusivity and permeation of CBD across excised colorectal membrane compared with pristine CBD. Furthermore, in vitro release studies demonstrated a trend of stable CBD release kinetics with congruent absorptivity as per the diffusion study. Considering these outcomes, the nano-sized CBD-loaded transferosomes can be embedded within a colloidal enema or suppository to extend its use in patients under palliative care with stable rectal absorption. The formulated CBD-loaded transfersomes could easily be combined into a suppository base and further optimization could be carried out to control the release and onset of action. Thereafter in vivo testing using a suitable CBD control should be carried out to fully demonstrate the benefit of the proposed lipid nanocarriers for CBD.

## Figures and Tables

**Figure 1 pharmaceutics-14-00703-f001:**
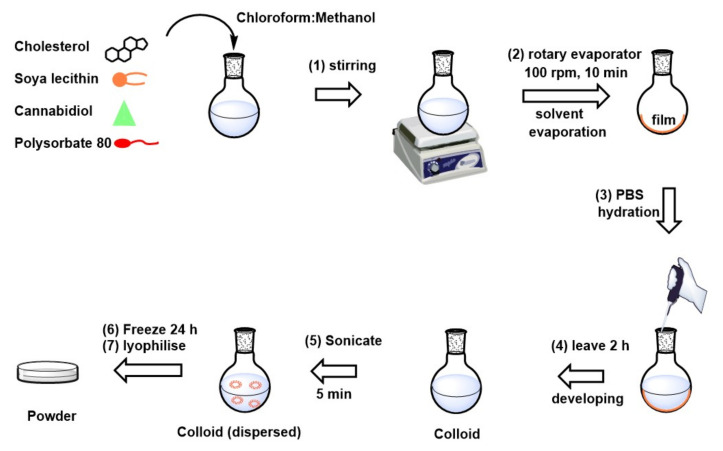
Schematic representation of methodology employed for the preparation of CBD-loaded transfersomes.

**Figure 2 pharmaceutics-14-00703-f002:**
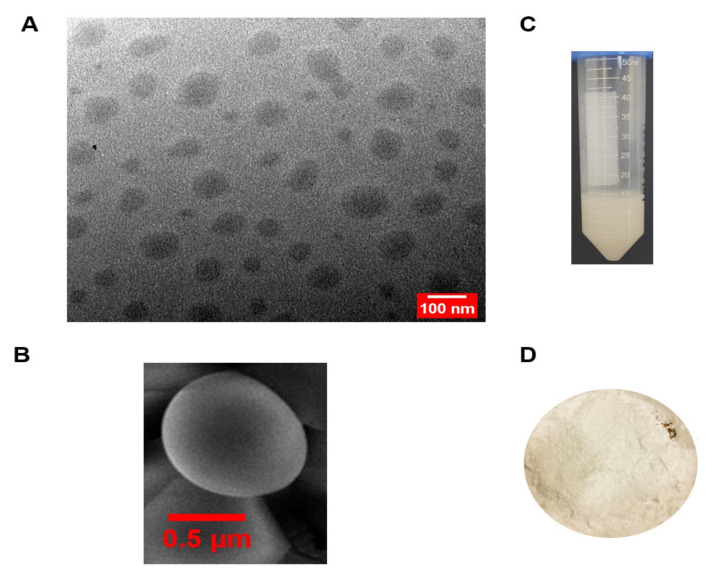
Typical macro- and microscopic imagery of the optimized CBD-loaded transfersome formulation (T6). (**A**) TEM image (scale bar 100 nm); (**B**) SEM image; (**C**) physical appearance of the transfersomes (T6); (**D**) lyophilized free-flowing powder of the transferomes (T6).

**Figure 3 pharmaceutics-14-00703-f003:**
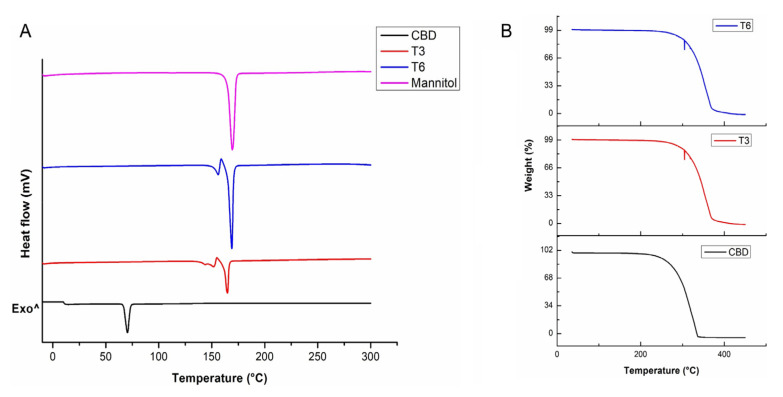
(**A**) DSC thermogram and (**B**) TGA thermogram of pure CBD, transfersomes (T3) and CBD-loaded transfersomes (T6) respectively.

**Figure 4 pharmaceutics-14-00703-f004:**
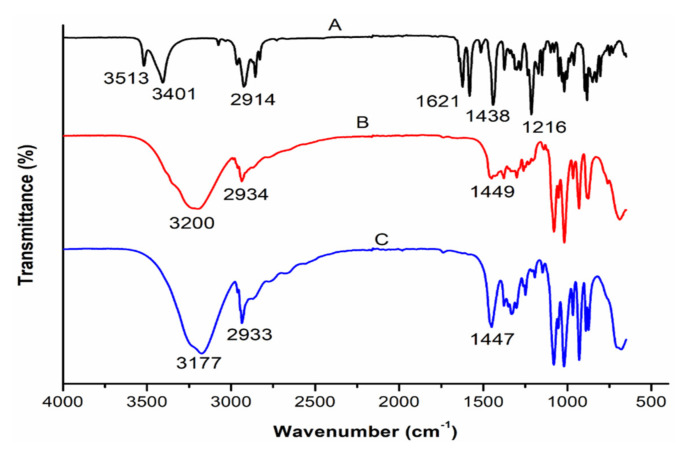
FTIR spectra of (A): pristine CBD; (B): CBD-free transferosomes; and (C): optimized CBD-loaded transfersomes (T6).

**Figure 5 pharmaceutics-14-00703-f005:**
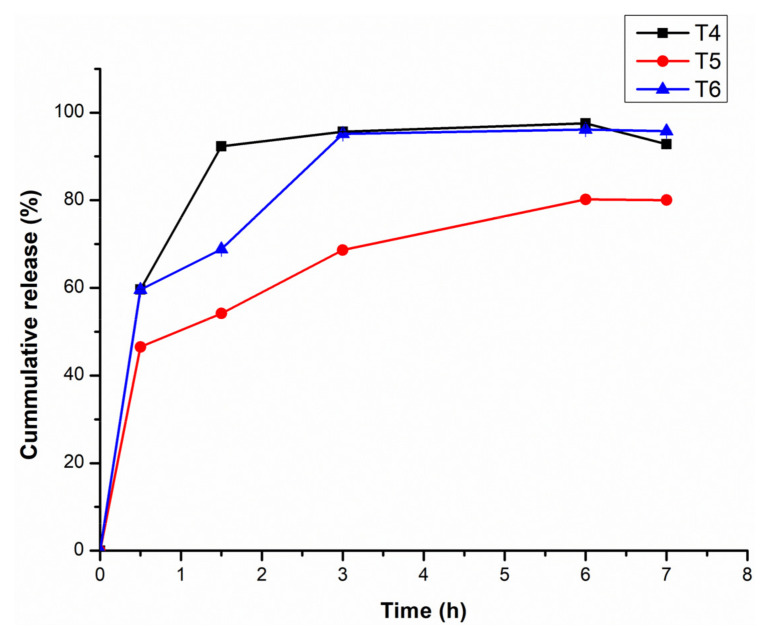
Cumulative release profile of CBD from the transferosomes over a period of 7 h at 37 °C. (*n* = 3) for T4: CBD-loaded transfersomes with 25 mg polysorbate 80, T5: CBD-loaded transfersomes with 50 mg polysorbate 80, and T6: CBD-loaded transfersomes with 100 mg polysorbate 80.

**Figure 6 pharmaceutics-14-00703-f006:**
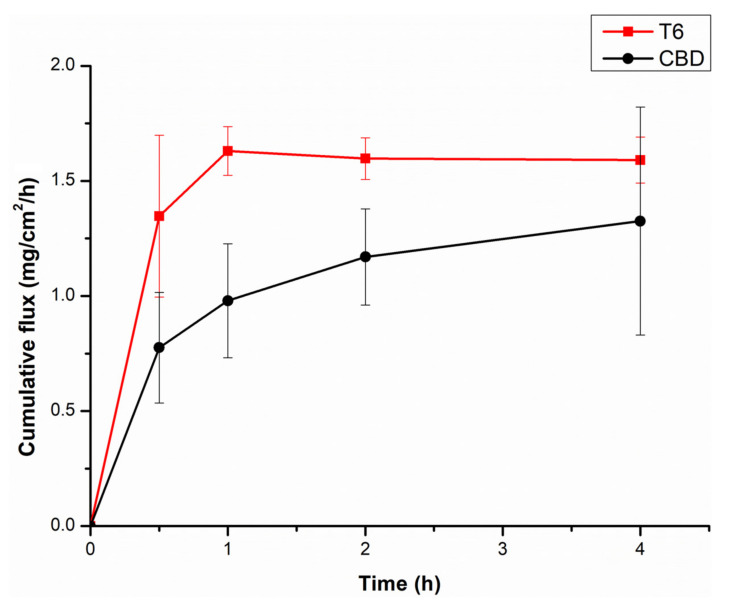
Cumulative permeation of pristine CBD and optimized CBD-loaded transferosomes across excised colorectal membrane (T6) over a period of 7 h at 37 °C. (*n* = 3).

**Figure 7 pharmaceutics-14-00703-f007:**
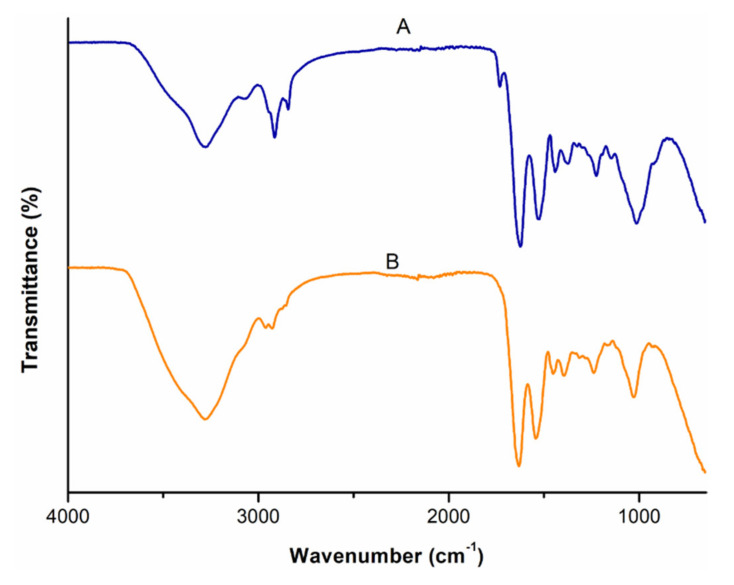
Changes in the FTIR spectra of the colorectal membrane recorded: (A): before permeation studies and (B): at the end of the permeation study.

**Figure 8 pharmaceutics-14-00703-f008:**
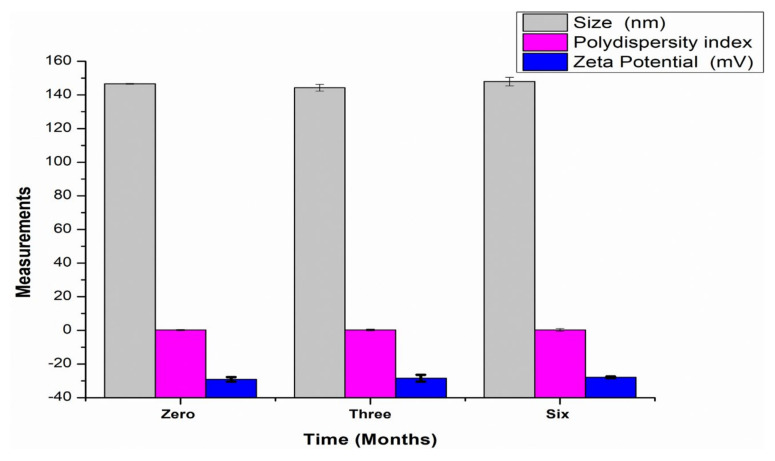
The graphical presentation of stability of transfersomes colloid for a period of six months based of measurement of the particle size, polydispersity index, and zeta potential at room temperature (25 °C) and ambient humidity.

**Figure 9 pharmaceutics-14-00703-f009:**
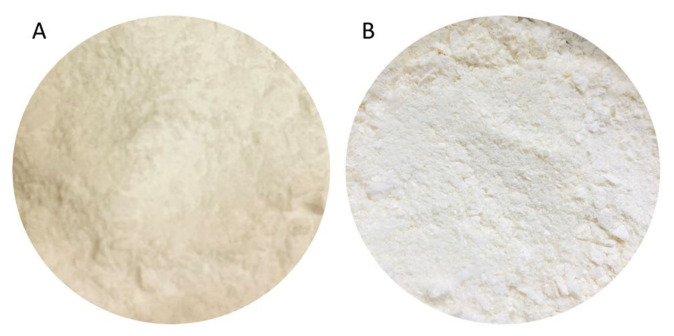
Images of the physical appearance of the transfersomes at (**A**): day 0 and (**B**): 6 months stored at room temperature (25 °C and at ambient humidity).

**Table 1 pharmaceutics-14-00703-t001:** Composition of lipids and solvents used for the formulation of CBD-loaded transfersomes.

Formulation ^1^	CBD (mg)	Polysorbate 80
T1	-	25
T2	-	50
T3	-	100
T4	50	25
T5	50	50
T6	50	100

^1^ All formulations contained soybean lecithin: 60 mg and cholesterol: 30 mg.

**Table 2 pharmaceutics-14-00703-t002:** Micomeritic and stability characterization of the CBD-loaded transferosomes.

Formulation	EE%	Size (nm) and PDI	Charge (mV)
**Transfersomes Colloid**			
T1		73.2 ± 0.065 & 0.24 ± 0.0050	−12.47 ± 0.38
T2	-	86.8 ± 0.76 & 0.25 ± 0.0050	−15.17 ± 1.36
T3	-	83.1 ± 0.39 & 0.30 ± 0.0035	−20.80 ± 0.95
T4	55.7 ± 0.19	102.2 ± 0.72 & 0.262 ± 0.012	−8.53 ± 2.00
T5	77.6 ± 0.14	124.0 ± 1.18 & 0.27 ± 0.0021	−9.20 ± 0.29
T6	80.0 ± 0.077	130.1 ± 0.64 & 0.29 ± 0.0056	−15.97 ± 1.30
CBD	-	2709 ± 1.0 & 0.504 ± 0.021	−4.9 ± 0.15
**Lyophilized Transfersomes Powder**			
T1	-	92.5 ± 0.77 & 0.26 ± 0.0047	−21.53 ± 0.61
T2	-	108.6 ± 0.80 & 0.30 ± 0.043	−27.40 ± 1.37
T3	-	101.5 ± 0.20 & 0.31 ± 0.030	−30.23 ± 0.71
T4	52.7 ± 0.56	121.7 ± 2.11 & 0.24 ± 0.0058	−20.37 ± 1.44
T5	75.6 ± 0.54	130.0 ± 1.65 & 0.22 ± 0.012	−25.30 ± 1.44
T6	78.6 ± 0.61	146.6 ± 0.23 & 0.25 ± 0.0051	−29.13 ± 0.40

**Table 3 pharmaceutics-14-00703-t003:** In vitro kinetic modeling of CBD release from the transfersomes.

Formulation	Zero Order Model (R^2^)	First Order Model	Higuchi Model (R^2^)	Korsmeyer-Peppas Model
T4	0.4997	0.4211	0.6716	0.1364
T5	0.9333	0.7282	0.8265	0.2254
T6	0.7292	0.7342	0.7697	0.1902

## Data Availability

The data presented in this study is available on request from the corresponding author.

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
