# Peer review of "Development of Stable Nano-Sized Transfersomes as a Rectal Colloid for Enhanced Delivery of Cannabidiol"

_pharmaceutics, 2022, doi:10.3390/pharmaceutics14040703_

Round 1

Reviewer 1 Report

The manuscript entitled “Development of stable nano-sized transfersomes as a rectal colloid for enhanced delivery of cannabidiol” presented research study on preparation and characterization of three formulation with difference in amount of polysorbate 80. In this study, authors reported the formation of stable transfersomes with higher rectal tissue penetration. However, this manuscript needs to be extensively revised before publication. Some suggestions/comments are as follows:

  1. In vitro experiments to demonstrate enhanced permeation need to be added.
  2. Was there any difference in particle size and PDI of transfersomes before and after lyophilization, as measured by Malvern zeta sizer. Please explain the findings.
  3. Table 2, some PDI results contain ±SD and some are missing. Please make it uniform.
  4. Table 2, zeta potential values for T5 and T6 are reported in the wrong row.
  5. Line 369, formulation T4 and T5 or T4 and T6?
  6. Line 379, is T3 the correct formulation. T3 do not contain CBD as per the formulation table.
  7. For in vitro release, authors need to include CBD solution as control group? Drug release results for formulation T5 should be re-explained with proper reason.
  8. Table 3, please check the formulation code (T1, T2, T3 or T4, T5, T6).
  9. Did the authors study storage stability in terms of drug content?
  10. Please check abbreviations, reference, and typo errors before resubmission.

Author Response

Thank you for the valuable comments.

Point-by-point response to the reviewer’s comments as been uploaded.

Reviewer 2 Report

The article entitled Development of Stable Nano-Sized Transfersomes as a Rectal

 Colloid for Enhanced Delivery of Cannabidiol is a document of interesting subject matter.

However, it needs some major changes before being accepted. Make the following corrections:

  1. Authors should add a schematic regarding preparation of nano-sized Transfersomes including drug, for clarity.
  2. Conclusion is short, needs to be expanded mentioning the application of the methods developed and use- further future extension of the work.
  3. Some of results lack stastical significance. Authors are advised to mention.
  4. In this work, nano-sized Transfersomes containing drug synthesized. In other words, our samples are liquid samples and not solid samples. With this in mind, authors how to get the size by an ordinary TEM tool due high vacuum. Please more clarify in this case.

For characterization the exact size of liquid samples, it is needed to having Cryo-TEM.

  1. Please try to more discussion on relationship between size and behavior of drug release
  2. Why did the authors choose Soybean lecithin (SL), polysorbate 80,Triton X-100 as surfactant and surfactantco- for delivery of the drug?
  3. The authors should cite and discuss some related studies about these nano-sized Transfersomes especially in EE% and In-vitro release.
  4. The conclusion is a bit too concise. Please make a general conclusion of the study.
  5. Please try to compare the results of your paper with another similar study.
  6. The results in the Table 2 confirm  instability of   some of the CBD-loaded transferosomes. Please input more discussion on this issue and its effect on release profile.
  7. Please improve introduction by introducing current challenges by standard medical to treat colorectal cancer and then introducing nanotechnology to combat with current challenges. Please cite to the paper in following:
  • DOI: 10.1007/s11051-020-05129-6
  • DOI: 10.1016/j.ijbiomac.2021.12.052
  • DOI: 10.1007/s12247-022-09621-5
  • DOI: 10.1016/j.jddst.2022.103138
  • DOI: 10.3390/app12010477
  • DOI: 10.3390/pharmaceutics14030472

Author Response

Thank you for the valuable and critical comments.

Point-by-point response to the reviewer’s comments as been uploaded.

Reviewer 3 Report

The article by Moqejwa et al studies the rectal administration of CNB ex vivo, the introduction is well argued justifying the reason for the interest of this research. The results are consistent with the discussion. However, some points need to be clarified:

1. The preparation of the vesicles needs to be better explained, how do they not rejoin each other, for example, by leaving them in suspension? Also, what material do you use to filter them? L140.
2. TGA explanation should be improved, even, the authors could illustrate the %.
3. Table 2: How the authors measure a size of 1085 nm for CBD and only 130 n for formulations, this is ok? Please explain. The charge data for T5 is also missing... Please check. Even, how the charge of CBD, which is negative, reduced the charge of formulations? The pKa should not affected if the samples is in the same medium
4. Figure 4. Adding the error bars.
5. Figure 7. must be split in two in order to see the graphs properly. Furthermore, above this stability point, the authors should check the stability of the drug, not of the formulation (it could have undergone some intramolecular variation).

Author Response

Thank you for the excellent comments.

Point-by-point response to the reviewer’s comments as been uploaded.

Round 2

Reviewer 1 Report

Authors answered to each queries of the reviewer. Manuscript can be accepted for publication in the present form.

Reviewer 2 Report

Dear authors,

Many thanks for your correction.

Authors addressed all comments carefully.

So, my suggestion is "Accept".

Reviewer 3 Report

The authors have satisfactorily answered the questions. The article can be considered for publication